# *Orf Virus*-Based Vaccine Vector D1701-V Induces Strong CD8+ T Cell Response against the Transgene but Not against ORFV-Derived Epitopes

**DOI:** 10.3390/vaccines8020295

**Published:** 2020-06-10

**Authors:** Alena Reguzova, Michael Ghosh, Melanie Müller, Hanns-Joachim Rziha, Ralf Amann

**Affiliations:** Department of Immunology, Interfaculty Institute for Cell Biology, University of Tübingen, 72076 Tübingen, Germany; alena.reguzova@uni-tuebingen.de (A.R.); Michael.ghosh@uni-tuebingen.de (M.G.); melanie.mueller@uni-tuebingen.de (M.M.); achim.rziha@ifiz.uni-tuebingen.de (H.-J.R.)

**Keywords:** *Orf virus*, ORFV, *Parapoxvirus*, viral vector, CD8+ T cells, epitopes, MHC class I, mass spectrometry, immunopeptidome, immune response, vaccine

## Abstract

The potency of viral vector-based vaccines depends on their ability to induce strong transgene-specific immune response without triggering anti-vector immunity. Previously, *Orf virus* (ORFV, *Parapoxvirus*) strain D1701-V was reported as a novel vector mediating protection against viral infections. The short-lived ORFV-specific immune response and the absence of virus neutralizing antibodies enables repeated immunizations and enhancement of humoral immune responses against the inserted antigens. However, only limited information exists about the D1701-V induced cellular immunity. In this study we employed major histocompatibility complex (MHC) ligandomics and immunogenicity analysis to identify ORFV-specific epitopes. Using liquid chromatography-tandem mass spectrometry we detected 36 ORFV-derived MHC I peptides, originating from various proteins. Stimulated splenocytes from ORFV-immunized mice did not exhibit specific CD8+ T cell responses against the tested peptides. In contrast, immunization with ovalbumin-expressing ORFV recombinant elicited strong SIINFEKL-specific CD8+ T lymphocyte response. In conclusion, our data indicate that cellular immunity to the ORFV vector is negligible, while strong CD8+ T cell response is induced against the inserted transgene. These results further emphasize the ORFV strain D1701-V as an attractive vector for vaccine development. Moreover, the presented experiments describe prerequisites for the selection of T cell epitopes exploitable for generation of ORFV-based vaccines by reverse genetics.

## 1. Introduction

Induction of strong transgene-specific immune responses and negligible anti-vector immunity are the prerequisites of efficient viral vector-based vaccines [1,2,3]. Recently, *Orf virus* (ORFV, *Parapoxvirus*) was reported as a novel viral vector for the expression of various foreign antigens. The very restricted host range (sheep and goats) and the exceptionally strong immune-modulating properties make ORFV a promising viral vector candidate [4,5,6,7,8,9]. Importantly, different derivatives of the Vero cell culture adapted vector virus strain D1701-V were apathogenic even after high dose infection of immunosuppressed sheep [10]. Whether and which of the ORFV genes deleted in D1701-V remains to be clarified [11].

The apathogenic Vero cell culture-adapted ORFV strain D1701-V has been used to generate recombinants by substituting viral genes with the foreign genes, which led to further attenuation [5,10,12]. ORFV-based vaccines have mediated protective immunity against different viral infections, such as rabbit hemorrhagic disease virus [13], classical swine fever [14], Borna disease virus [15], pseudorabies virus [12,16,17], rabies virus [18] and influenza A virus [19]. Lately, the therapeutic application of ORFV vector-based vaccine against tumours induced by cottontail rabbit papillomavirus has been described [20]. The major advantage of using recombinant ORFV for vaccination is based on the fact that ORFV in general elicits only a short-lived virus-specific immunity in its natural hosts allowing frequent reinfections due to the absence of virus neutralizing antibodies [4,5,21,22,23,24,25,26,27]. In turn, this feature enables repeated immunizations using ORFV recombinants to enhance humoral immune responses against inserted antigens [13,15,16,17,18,19]. Analyses of human patients accidentally infected with wild-type ORFV, whose lesions generally resolve in short times, suspected a human immune response very similar to that of sheep [28]. However, it is not known whether a vaccination using D1701-V ORFV induces ORFV-specific T lymphocyte responses.

Induced vector virus-specific T cell responses can limit the immunogenicity of vector-encoded transgenes [2,29]. Some studies indicate that cellular immunity against viral epitopes prevents efficient priming of T cell responses to the antigens delivered by recombinant viruses [30,31,32]. Furthermore, cytotoxic T lymphocyte (CTL) responses to immunodominant epitopes of vector proteins have been shown to suppress CD8+ T cell priming to subdominant epitopes of the transgene [29]. While increased numbers of activated CTLs have been found after *Orf virus* reinfection at the site of lesion in sheep [22], CD8+ T cells did not appear to be essential for viral clearance later in the infection [21]. Hence, we assume that along with the lack of specific ORFV-neutralizing antibodies, cellular immunity against D1701-V ORFV vector is also negligible.

Understanding the specificity of anti-vector responses requires evaluation of both the epitopes presented and the corresponding CD8+ T cells. For this, virus-derived major histocompatibility complex (MHC) class I restricted peptides have to be identified and any elicited CD8+ T lymphocytes detected and quantified. For small viruses, overlapping libraries of synthetic peptides can be used for the epitope discovery [33,34], while this technique is not applicable for large viruses of the poxvirus family including ORFV. Instead, epitope search for those viruses is mostly based on *in silico* prediction of the MHC I-bound peptides [35,36,37]. However, many of these peptides might not be of physiological relevance if they are not presented on the cells during infection [36,38]. Thus, the identification of specific MHC-associated peptides, or immunopeptidome, which are naturally processed and presented by the virus infected cells employing mass spectrometry has become a feasible alternative [38,39,40,41,42]. For example, by using this approach 73 H-2K^b^ and 97 H-2D^b^ vaccinia virus (VACV)-derived peptides have been described for murine MHC I molecules [43], as well as 10 and 64 peptides for human leukocyte antigen (HLA)-A2 and B7, respectively [44]. For the modified vaccinia virus Ankara (MVA), 98 unique HLA class I associated peptides have been published [40].

In this study we report for the first time the identification of ORFV-specific epitopes in a combined approach of MHC ligandomics and immunogenicity analysis. Using liquid chromatography-tandem mass spectrometry (LC-MS/MS) and database annotation we detected 36 peptides as ligands for mouse MHC class I allele H-2K^b^, originating from various ORFV proteins. Immunogenicity of the identified peptides was evaluated in mice after two times administration of ORFV recombinants. We demonstrate that D1701-V ORFV does not induce CD8+ T cell responses against identified virus-derived MHC class I restricted peptides, but a strong CTL immune response directed against the encoded transgene.

## 2. Materials and Methods

### 2.1. Cells and Viruses

HeLa cells transfected with a mouse MHC class I gene H-2K^b^ (HeLa-K^b^ cells) were obtained from the cell line bank of the Department of Immunology, University of Tübingen, Germany and maintained in RPMI (Life Technologies, Carlsbad, CA, USA) supplemented with 10% fetal bovine serum (FBS) (Capricorn Scientific, Ebsdorfergrund, Hessen, Germany), 50 U/mL Penicillin and 50 μg/mL Streptomycin (Sigma-Aldrich, St Louis, MO, USA) as described previously [45]. Splenocytes from immunized mice were cultured in RPMI (Life Technologies, Carlsbad, CA, USA) supplemented with 10% FBS (Capricorn Scientific, Ebsdorfergrund, Hessen, Germany), 50 U/mL Penicillin and 50 μg/mL Streptomycin (Sigma-Aldrich, St Louis, MO, USA). D1701-V-D12-mCherry ORFV (abbreviated as V-D12-mCherry) was described previously [11]. The mouse ovalbumin (Ova) gene was synthesized (Gene Art, Thermo Fisher Scientific, Waltham, MA, USA) and cloned as a *Hind*III-*EcoR*I fragment into plasmid pV12-mCherry [11]. The resulting transfer plasmid pV12-mOva was used for transfection of Vero cells infected with D1701-V12-mCherry-D12-GFP [11] to replace the mCherry gene by the Ova gene. Ova-specific PCR allowed selection and purification of the Ova-expressing recombinant D1701-V12-Ova-D12-GFP essentially as described earlier [11]. Virus stocks were purified from clarified supernatant of infected cells followed by ultracentrifugation through 36% sucrose cushion, and virus titers were determined by standard plaque assay.

### 2.2. Infection of Cells

For flow cytometric analysis, 10^5^ cells were seeded into 24-well plate (Greiner CELLSTAR, Kremsmünster, Austria) in 0.5 mL medium and infected by adding appropriate volume of indicated virus to reach the required multiplicity of infection (MOI). For the isolation of ORFV-derived H-2K^b^-presented peptides 10^8^ HeLa-K^b^ cells were infected with V-D12-mCherry (MOI = 5) in 10 mL of medium. Two hours post infection, the cell concentration was adjusted to 10^6^ per ml and cells were cultured for additional 18 h. After harvesting, cells were washed twice with phosphate buffer saline (PBS) and the cell pellet was frozen at −80 °C.

### 2.3. Flow Cytometry Analysis of Infected Cells

Cell viability of HeLa-K^b^ cells 20 h post infection was determined by Zombie Aqua staining (BioLegend, San Diego, CA, USA) followed by flow cytometry. The infection rate was calculated by flow cytometric detection of mCherry or Green Fluorescent Protein (GFP) expressing cells. Quantification of H-2K^b^ molecules on the surface of ORFV-infected HeLa-K^b^ cells was done using QIFIKIT (Dako, Glostrup, Denmark) according to the manufacturer’s protocol. Detection of Ova_257-264_ SIINFEKL peptide on the surface of ORFV-infected cells was performed after staining with Allophycocyanin (APC) anti-mouse H-2K^b^ bound to SIINFEKL antibody (BioLegend, San Diego, CA, USA; clone 25-D1.16). Samples were acquired on a BD Fortessa flow cytometer (BD Biosciences, San Jose, CA, USA) and data were analyzed using FlowJo software version 10 (Tree Star Inc., Ashland, OR, USA).

### 2.4. Isolation of MHC I-Bound Peptides and Mass Spectrometric Analysis

MHC I-bound peptides were isolated using standard immunoaffinity purification [46,47]. Snap-frozen ORFV-infected HeLa-K^b^ cells were lysed in 10 mM CHAPS/PBS (AppliChem, St. Louis, MO, USA/Gibco, Carlsbad, CA, USA) with 1× protease inhibitor (Complete; Roche, Basel, Switzerland). H-2K^b^-presented peptides were isolated using monoclonal antibody directed against anti-mouse H-2K^b^ (clone Y-3, produced in-house) covalently linked to CNBr-activated Sepharose (GE Healthcare, Chalfont St Giles, UK). MHC–peptide complexes were eluted by repeated addition of 0.2% trifluoroacetic acid (TFA) ( Merck, Whitehouse Station, NJ, USA). Eluted MHC ligands were purified by ultrafiltration using centrifugal filter units (Amicon; Millipore, Burlington, MA, USA). Peptides were desalted using ZipTip C18 pipette tips (Millipore, Billerica, MA, USA), eluted in 35 μL 80% acetonitrile (Merck, Darmstadt, Germany) containing 0.2% TFA, vacuum-centrifuged and resuspended in 25 μL of 1% acetonitrile with 0.05% TFA (Merck, Whitehouse Station, NJ, USA). Samples were stored at −20 °C until LC-MS/MS analysis. For LC-MS/MS analysis peptides were separated by reversed-phase liquid chromatography (nano-UHPLC, UltiMate 3000 RSLCnano; Thermo Fisher Scientific, Waltham, MA, USA) and analyzed in an online coupled Orbitrap Fusion Lumos mass spectrometer (Thermo Fisher Scientific, Waltham, MA, USA). Samples were analyzed in triplicates and sample shares of 33% trapped on a 75 μm × 2 cm trapping column (Acclaim PepMap RSLC; Thermo Fisher Scientific, Waltham, MA, USA) at 4 μL/min for 5.75 min. Peptide separation was performed at 50 °C and a flow rate of 175 nl/min on a 50 μm × 25 cm separation column (Acclaim PepMap RSLC; Thermo Fisher Scientific, Waltham, MA, USA) applying a gradient ranging from 2.4–32.0% of acetonitrile over the course of 90 min. Samples were analyzed on the Orbitrap Fusion Lumos implementing a top-speed collision-induced dissociation method with survey scans at 120,000 (120 k) resolution and fragment detection in the Orbitrap at 60,000 (60 k) resolution. The mass range was limited to 400–650 m/z with precursors of charge states 2+ and 3+ eligible for fragmentation.

### 2.5. Database Search and Filtering

LC-MS/MS results were processed using Proteome Discoverer (v.1.3; Thermo Fisher Scientific, Waltham, MA, USA) and database search was performed with the Sequest HT search engine (University of Washington, Seattle, WA, USA). The human proteome was used as reference database annotated by the UniProtKB/Swiss-Prot, status 27 September 2013 containing 20,279 reviewed sequences plus the ORFV proteome [48]. The search combined data of three technical replicates was not restricted by enzymatic specificity and oxidation of methionine residues was allowed as dynamic modification. Precursor mass tolerance was set to 5 ppm, and fragment mass tolerance to 0.02 Da. False discovery rate was estimated using the Percolator node [49] and was limited to 5%. For MHC class I ligands, peptide lengths were limited to 8–12 amino acids. MHC annotation of peptides fitting to the H-2K^b^ peptide motif was performed using NetMHCpan4 database [50,51].

### 2.6. Peptide Synthesis

The automated peptide synthesizer Liberty Blue (CEM, Charlotte, North Carolina, USA) was used to synthesize peptides by the 9-fluorenylmethyl-oxycarbonyl/tert-butyl (Fmoc/tBu) strategy. Purity of the peptides was assessed by reversed-phase liquid chromatography (nano-UHPLC, UltiMate 3000 RSLCnano, Thermo Fisher Scientific, Waltham, MA, USA) coupled online to LTQ Orbitrap XL (Thermo Fisher Scientific, Waltham, MA, USA). From the 36 identified ORFV-derived peptides 32 were synthesized (1–32 indicated in Table 1), while peptides 33–36 (Table 1) could not be obtained due to unsuccessful synthesis.

### 2.7. Immunization of Mice

Female C57BL/6 mice (8–12 weeks old) were obtained from Jackson Laboratories (Jackson Labs, Bar Harbor, ME, USA) and were housed in the biosafety level 1 animal facility at the University of Tübingen, Germany. All animals were handled in strict accordance with good animal practice and complied with the guidelines of the local animal experimentation and ethics committee—experiments were conducted under Project License (Nr. IM 1/15). Mice were immunized intramuscularly (i.m.) in the anterior tibialis with 7.5 × 10^6^ plaque forming units (PFU) of V12-Ova-D12-GFP or control V-D12-mCherry ORFV recombinant. After 10 days mice received a homologous i.m. boost immunization with 10^7^ PFU of V12-Ova-D12-GFP or V-D12-mCherry. Mice were sacrificed one week after the second administration, and the splenocytes were used for immunological testing as described below.

### 2.8. IFN-γ ELISPOT Assay

The ELISPOT assay was performed using the mouse IFN-ƴ ELISPOT kit (Mabtech, Nacka Strand, Schweden). Immune splenocytes were tested either separately from individual mice or pooled as described in the “Results” section. In total 2 × 10^5^ splenocytes per well were either stimulated with peptide pools (10 µg/mL of each peptide) or with Ova_257-264_ SIINFEKL peptide (1 µg/mL) for 21 h. Each peptide pool contained eight randomly mixed synthetic peptides, corresponding to the identified ORFV-derived ligands (numbers 1–32 as listed in Table 1). Phytohemagglutinin (PHA-L) (Roche, Basel, Switzerland) (10 µg/mL) was used as a positive control, unstimulated cells served as a negative control. Developed spots were automatically counted using an ImmunoSpot S5 analyzer (Cellular Technology Limited, Cleveland, OH, USA) and ImmunoSpot software (Cellular Technology Limited, Cleveland, OH, USA). Responses were considered to be positive when the mean spot count per well was at least threefold higher than the mean number of spots in negative control wells. Overlapping spots in wells with strong IFN-ƴ response hamper the detection of reliable counts, therefore spot counts of >1.000 per well were set to 1.000.

### 2.9. MHC Dextramer Staining

Splenocytes of individual mice were stained with H-2K^b^ Ova_257-264_ dextramer-PE (Immudex, Kopenhagen, Denmark) according to the manufacturer’s protocol. Afterwards, cells surface staining was performed with anti-CD90.2-APC (BioLegend, San Diego, CA, USA; clone 53-2.1), anti-CD8-Alexa 700 (BioLegend, San Diego, CA, USA; clone 53-6.7) and anti-CD19-FITC (BioLegend, San Diego, CA, USA; clone 6D5) antibodies for 30 min at 4 °C. Cells were washed twice with PBS and viability staining was performed using Zombie Aqua (BioLegend, San Diego, CA, USA) according to the manufacturer’s instructions. Samples were acquired on a BD Fortessa flow cytometer (BD Biosciences, San Jose, CA, USA) and data were analyzed using FlowJo software version 10 (Tree Star Inc., Ashland, OR, USA).

### 2.10. Intracellular Cytokine Staining (ICS)

Splenocytes were tested either separately from individual mice or pooled as described in the “Results” section; 2 × 10^6^ cells per well were seeded into 96-well plate (Corning, NY, USA) and stimulated for 16 h with individual ORFV-derived peptides (10 μg/mL), with 1 µg/mL of Ova_257-264_ SIINFEKL peptide, or with control ORFV at MOI 1 in the presence of brefeldin A (Sigma-Aldrich, St Louis, MO, USA), GolgiStop (BD Biosciences, San Jose, CA, USA) and anti-CD107a-FITC antibody (BioLegend, San Diego, CA, USA; clone 1D4B). PHA-L-stimulated cells (10 µg/mL) served as a positive control, unstimulated cells were used as a negative control. After incubation, cell viability staining was performed using Zombie Aqua (BioLegend, San Diego, CA, USA) followed by cell surface staining with anti-CD90.2-APC (BioLegend, San Diego, CA, USA; clone 53-2.1), anti-CD8-Alexa 700 (BioLegend, San Diego, CA, USA; clone 53-6.7) and anti-CD4-Pacific Blue (BioLegend, San Diego, CA, USA; clone RM4-5) antibodies for 30 min at 4 °C. Afterwards, cells were fixed and permeabilized using BD Cytofix/Cytoperm (BD Biosciences, San Jose, CA, USA) for 30 min at 4 °C and incubated with anti-TNFa-PE-Cy7 (BioLegend, San Diego, CA, USA; clone MP6-XT22), anti-IFNg-BV711 (BioLegend, San Diego, CA, USA; clone XMG1.2) and anti-IL-2-PE (BioLegend, San Diego, CA, USA; clone JES6-5H4) antibodies for 30 min at 4 °C. Flow cytometric measurements and data analysis were performed as described above.

### 2.11. In Vitro Re-Stimulation of Splenocytes

In total 2 × 10^6^ pooled splenocytes from V12-Ova-D12-GFP or from control V-D12-mCherry immunized mice per well were seeded into a 24-well plate (Greiner CELLSTAR, Kremsmünster, Austria). Cells were re-stimulated either with 10 µg/mL of the individual synthetic peptides representing identified ORFV-derived epitopes or with 1 µg/mL of Ova_257-264_ SIINFEKL peptide. Unstimulated cells were used as a negative control. IL-2 was added on day 2 and 5 after re-stimulation at a final concentration of 20 U/mL. On day 7, splenocytes were re-stimulated with 4 × 10^6^ feeder cells per well (200 Gy irradiated splenocytes from syngeneic nonimmune mice) and loaded with 10 µg/mL of individual synthetic peptides. IL-2 was added on days 9 and 12 as indicated above. After 14 days cells were harvested, and ICS was performed as previously described.

### 2.12. Statistical Analysis

Statistical significance was examined with GraphPad Prism 4.02 software (GraphPad, San Diego, CA, USA). For comparison between two conditions or groups, an unpaired student’s *t*-test was used to determine significance. A value of *p* < 0.05 was considered significantly different.

## 3. Results

### 3.1. ORFV Vector Strain D1701-V Efficiently Induces Transgene-Specific CD8+ T Cell Response

To date, the induction of CD8+ T cell responses by ORFV strain D1701-V has not been analyzed in detail. In order to test whether a homologous immunization regimen with recombinant D1701-V ORFV elicits a specific CD8+ T cell response to the vectored antigen, V12-Ova-D12-GFP encoding Ova was injected to C57BL/6 mice (H-2K^b^ positive) twice by i.m. route. For negative control mice were immunized with the control recombinant V-D12-mCherry. The immune response against the H-2K^b^-restricted CD8+ T cell epitope SIINFEKL was measured in splenocytes one week after the second immunization

We observed that V12-Ova-D12-GFP administration elicited a strong Ova-specific CD8+ T cell response. *Ex vivo* quantification of CTLs by H-2K^b^ Ova_257-264_ dextramer staining showed a high frequency of 42.9% specific CD8+ T cells (Figure 1A). The functionality of Ova-specific CD8+ T lymphocytes was measured by production of the pro-inflammatory cytokines interferon-gamma (IFN-γ), tumor necrosis factor alpha (TNF-α) and interleukin-2 (IL-2), as well as by the expression of lysosomal-associated membrane protein 1 (LAMP-1) known as CD107a. The results revealed that IFN-ƴ was expressed in 52.9%, TNF-α in 51.0%, IL-2 in 13.7% and CD107a in 59.3% of CD8+ T cells (Figure 1B), whereas no Ova-specific response was detected in mice immunized with negative control ORFV (Figure 1A,B). Notably, the CTL response against Ova-derived epitope was dominated by multifunctional CD8+ T cells producing simultaneously IFN-ƴ, TNF-α and CD107a (Figure 1C).

These results demonstrate that ORFV strain D1701-V mediated strong transgene-specific CD8+ T cell immunity in mice after two intramuscular immunizations, which was not induced by the vector virus without foreign immunogen.

### 3.2. HeLa-K^b^ Cells Are Suitable Targets for the Identification of ORFV-Derived Peptides

No ORFV-specific T cell epitopes have been identified so far. To generate a comprehensive inventory of ORFV-derived MHC class I restricted peptides that might be presented to CD8+ T cells during immunization of the H-2K^b^ positive C57BL/6 mice, HeLa cell line stably transfected with the mouse MHC class I allele H-2K^b^ (HeLa-K^b^ cells) was used. HeLa-K^b^ cells are easily expandable in cell culture and exhibited high infection efficiency with ORFV in combination with high cell viability (Figure 2A). For infection the V-D12-mCherry virus expressing mCherry under control of the early eP2 promoter was used, which allows early detection of infected cells without the production of viral infectious progeny [11]. Twenty hours after infection (MOI 5), 48.1% of cells expressed mCherry (Figure 2A). Increasing the MOI up to 50 resulted in 87.1% of infected cells (Figure 2A). Notably, high cell viability of 95.3–90.2% was found after 20 h of infection with all tested MOI (Figure 2A).

Furthermore, we demonstrated that ORFV-infected HeLa-K^b^ cells expressed high levels of H-2K^b^ molecules. The absolute numbers of H-2K^b^ molecules on the HeLa-K^b^ cell surface were determined by flow cytometry using the quantitative indirect immunofluorescence kit. Thus, non-infected cells exhibited approximately 4 × 10^5^ H-2K^b^ molecules per cell (Figure 2B). Infection with the ORFV for 20 h with MOI 3 or 5 significantly increased H-2K^b^ surface expression up to 5.5×10^5^ molecules per cell (Figure 2B). Higher MOI (10–50) resulted in the absolute numbers of H-2K^b^ molecules from 4.2–3.6 × 10^5^ that were comparable with the non-infected cells (Figure 2B). These data indicate that infected HeLa-K^b^ cells provide sufficient amounts of H-2K^b^ molecules for the elution of ORFV ligands.

Moreover, we showed that HeLa-K^b^ cells enable efficient presentation of HeLa-K^b^ restricted peptide SIINFEKL after infection with V12-Ova-D12-GFP recombinant. Thus, a complex of H-2K^b^ molecules with SIINFEKL peptide was detected on the HeLa-K^b^ cells by flow cytometry 20 h post infection with MOI 5 (Figure 2C,D). Importantly, nearly all ORFV-infected cells exhibited H-2K^b^-SIINFEKL complexes on their surface (Figure 2C). No specific surface staining was observed after exposure to the control virus (Figure 2C,D). Fluorescence intensity of the H-2K^b^-SIINFEKL staining of V12-Ova-D12-GFP infected cells was even higher than of the cells, loaded with synthetic SIINFEKL peptide (Figure 2D). This can be explained by increased numbers of H-2K^b^ molecules on the cell surface after infection with MOI 5 as described above. The recombinant V12-Ova-D12-GFP virus might also express a higher amount of SIINFEKL peptide as compared to the restricted number of loaded peptide molecules.

Altogether, these results indicate the ability of HeLa-K^b^ cells to endogenously process ORFV expressed proteins and efficiently present H-2K^b^ restricted epitopes. Hence, HeLa-K^b^ cells were chosen as target cells for ORFV infection and subsequent MS analysis.

### 3.3. Identification of ORFV-Derived H-2K^b^-Presented Peptides by LC-MS/MS

HeLa-K^b^ cells were infected for 20 h with ORFV at MOI 5 and the eluted H-2K^b^-presented peptides were analyzed by LC-MS/MS. This approach resulted in identification of 1460 unique HeLa-K^b^-presented peptides, originating from both infected and non-infected cells. Binding prediction with NetMHCpan4 assigned 1294 (88.6%) peptides as ligands fitting to the H-2K^b^ peptide motif. In total 43 (3.3%) unique ORFV-derived HeLa-K^b^-presented peptides were identified. We further analyzed frequencies of individual amino acids in every epitope position for all 8–10-mer peptides considering the MHC anchor amino acid preferences of the H2-K^b^ allele. Finally, 36 (2.78%) unique peptides were selected as ORFV-derived H-2K^b^ ligands (Table 1).

### 3.4. Characteristics of ORFV-Derived H-2K^b^-Presented Peptides

The 36 ORFV-derived ligands are encoded by 23 (17.2%) out of 134 putative ORFV genes [48,52]. The majority—eight of the detected ligands (22.2%)—originated from the virion core protein, six peptides (16.7%) were derived from the RNA-polymerase subunit and five peptides (13.9%) emerged from the IMV membrane protein. Epitope occurrence from the various ORFV proteins suggests that all these proteins are available to the antigen presentation pathway. While we performed the ligandome analysis at only one time point after infection (20 h), different kinetic classes of viral gene expression were present among the source proteins of the identified epitopes [53,54].

We observed that in some proteins detected H-2K^b^-restricted ORFV ligands comprised overlapping peptide cores with extensions at either the amino- or carboxy-terminus (for example, ORF088 Virion core protein and IMV membrane protein, Table 1). In other cases, there were sets of non-overlapping peptides from the same protein (RNA-polymerase subunit protein, Table 1). From all identified ORFV-associated peptides 18 (50%) were 8-mers, 17 (47.2%) 9-mers and 1 (2.8%) was a 10-mer.

We next determined the binding affinity of these peptides to H-2K^b^ molecules using NetMHCpan4, which categorizes peptides into weak (affinity <500 nM, % rank <2) and strong (affinity <50 nM, % rank <0.5) binders. The analysis indicated that 7 (19.4%) and 21 (58.3%) ORFV-derived peptides were strong binders with an affinity of ≤50 nM or <0.5% rank respectively (Table 1). Among others, 24 (66.7%) peptides with an affinity ≤500 nM and 11 (30.6%) with % rank <2 were defined as weak binders (Table 1).

Overall, we report for the first time the identification of 36 ORFV-derived unique H-2K^b^-restricted epitopes using immunopeptidomics.

### 3.5. Characteristics of ORFV-Derived H-2K^b^-Presented Peptides

All ORFV ligands, identified from HeLa-K^b^ cells by LC-MS/MS analysis were further evaluated for their immunogenicity. C57BL/6 mice received two times administration of V12-Ova-D12-GFP or control ORFV recombinant by i.m. route. Memory T cell responses against synthetic peptide pools were analysed in splenocytes one week after the second immunization by IFN-γ ELISPOT assay. Responses were considered positive when the mean count of spot forming units (SFU) per well was at least threefold higher than the mean number SFUs in the negative control wells. We found that the administration of control or V12-Ova-D12-GFP ORFV recombinants induced mean number of 11.25 and 20.5 SFU in the negative control wells, respectively (Figure 3A). Stimulation with the peptide pools elicited mean responses ranging between 11 and 20.3 IFN-γ spots in the control immunized mice and 22.25–29 SFUs in the V12-Ova-D12-GFP vaccinated mice (Figure 3) indicating no positive response to any of the tested peptides. In contrast, the response against the Ova_257-264_ SIINFEKL peptide exceeded 1000 IFN-ƴ SFU after two times V12-Ova-D12-GFP administration (Figure 3A,B). PHA as a positive control induced high lymphocyte responses in both groups of mice, indicating a functional state of the analysed lymphocytes (Figure 3).

These results demonstrate the ability of the ORFV strain D1701-V to elicit high magnitude transgene-specific cellular responses in mice after immunization and suggest a lack of T cell immunity against the ORFV vector.

### 3.6. ICS Validates Absence of CD8+ T Cell Responses to the Individual ORFV-Derived Peptides

To exclude the possibility of competitive effects among the different ORFV epitopes within one peptide pool we additionally performed ICS after 16 h *ex vivo* stimulation of pooled splenocytes from V12-Ova-D12-GFP and control ORFV immunized mice with the individual peptides. We also speculated that ORFV-epitope specific CTLs might produce other cytokines than IFN-ƴ. For this reason, we evaluated CD8+ T cell responses in terms of IFN-ƴ, TNF- α, IL-2 and CD107a expression. Response was considered positive when the frequency of responding cells was three times the value of the unstimulated control, and the simultaneous production of at least two cytokines indicates the positive response. The results shown in Figure 4 revealed an absence of specific IFN-ƴ, TNF-α and IL-2 responses to all tested peptides independently on predicted binding level. Mice responded to peptides 19 and 25, derived from ORFV015 and ORFV056 RNA-polymerase subunit RPO147 respectively, with threefold higher numbers of CD107a expressing CD8+ T cells (Figure 4C,D). However, both detected responses were not statistically significant compared to the unstimulated control. Moreover, CD107a-producing CD8+ T cells were monofunctional (Figure 4C,D). Thus, according to selection criteria the observed responses were considered as negative. Moreover, no positive responses were found by the use of ORFV for the stimulation of splenocytes (Figure 4). As observed previously, strong multifunctional responses were detected against Ova_257-264_ SIINFEKL peptide and PHA stimulation (Figure 4), indicating a functional state of the analysed lymphocytes. Coactivation of CD4+ T cells was not observed after cell stimulation with any of the tested ORFV ligands.

We further note that the employed immunogenicity assays are subject to a threshold of detection. It can be expected that memory CD8+ T cells specific to identified H-2K^b^ ORFV-derived peptides may represent a very small fraction of the splenic lymphocyte population *ex vivo* and can be below detection limits. To confirm the absence of memory CD8+ T cells to ORFV-derived peptides in general and of peptide 19 and 25 in particular, we re-stimulated pooled splenocytes from the immunized mice during 14 days with the individual peptides as described in Materials and Methods. Whenever specific cells are present among the splenocytes, peptide re-stimulation would lead to their expansion excluding priming of naïve T cells. If no memory CD8+ T-cells have been re-activated and expanded, cytokine expression would be comparable to the unstimulated control. We performed this assay for all identified ORFV-derived peptides to exclude the possibility that low-frequency memory CD8+ T cells might have been below detection limits of *ex vivo* ICS (Figure 4). Again, the CTLs did not positively respond in the presence of any of the used peptides. Percentages of IFN-ƴ, TNF-α, IL-2 or CD107a producing CD8+ T-lymphocytes were close to the unstimulated control cells (Appendix A) However, different cell culture conditions as well as the protocols applied might explain variation in the percentages of cytokine expressing cells in *ex vivo* ICS and after 14 days re-stimulation.

Altogether, we confirmed absence of recognition of the tested ORFV-derived H-2K^b^-presented peptides identified in this study and induction of strong multifunctional CD8+ T cell response against inserted antigen.

## 4. Discussion

Virus-derived vectors, facilitating the induction of strong transgene-specific immune responses, offer advantages over traditional vaccine technologies. However, anti-vector immunity represents a key limitation of current recombinant viral delivery platforms [55,56]. The recognition of virus-derived peptides, presented by MHC class I complexes on the surface of infected cells to CD8+ T lymphocytes, plays an important role in mediating antiviral immune responses against the viral vector [57]. A comprehensive view of the specificity of the anti-vector responses requires evaluation of both the epitopes presented and the corresponding CD8+ T cells. In our approach we employed LC-MS/MS analysis for ORFV for the first time to profile a viral immunopeptidome on mice where detailed evaluation of immunogenicity is possible.

The lack of sufficient ORFV uptake and vigorous cell death after virus exposure did not allow the use of murine primary bone marrow-derived macrophages and dendritic cells as well as the murine macrophage cell lines J774 and RAW264.7 (unpublished data), although they would provide a larger set of ORFV-derived peptides other than K^b^ ligands. Therefore, we used a transgenic HeLa cell line encoding the murine H-2K^b^ gene as a model of MHC class I epitope processing and presentation of ORFV infected cells. The combination of very good ORFV infection efficiency, unaffected cell viability and high H-2K^b^ expression levels with efficient model peptide presentation validated HeLa-K^b^ cells as suitable targets for MS analysis. Detection of the peptide presentation other than SIINFEKL would be of interest, however no antibodies are available to reveal ORFV-derived epitopes in the context of H-2K^b^.

In total, we were able to identify 36 naturally presented ORFV-derived H-2K^b^ peptides from HeLa-K^b^ cells. All discovered peptides are reported for ORFV for the first time. We were focusing on the early ORFV gene expression until 20 h post infection based on the observation that major VACV immunogenic epitopes are originating from early expression gene products and late viral proteins were recognized by fewer CD8+ T cells [43,58]. Hence, we supposed that the majority of immunogenic peptides would be derived from the ORFV proteins expressed early after infection. Interestingly, we detected peptides from all kinetic classes of ORFV proteins. However, we cannot discriminate whether early presentation of ORFV-derived epitopes resulted directly from entering virion proteins or from newly synthesized gene products as reported for MVA [40].

Immunogenicity analysis of the identified ORFV-specific epitopes revealed that none of the tested H-2K^b^ peptides is immunogenic in mice. Lack of specific memory CD8+ T lymphocyte responses to ORFV is in accordance with the previous observation that CD8+ T cells are not essential for the virus clearance after reinfection [21]. While the cell-depletion study proposed a role of CD4+ T lymphocytes for the control of viral replication [21], duration of anti-ORFV immunity is short-lived [24]. Hence, our data prove that cellular immunity against ORFV strain D1701-V is negligible. It is important to mention that the epitope discovery is limited to the sensitivity of the LC-MS/MS method, which may lead to an inefficient detection of low-abundance MHC class I presented ORFV peptides. However, major CD8+ T cell responses were shown to be associated with the high affinity for MHC class I complexes, even among the peptides of sufficient affinity to be presented [43]. Other reasons could partly explain the absence of memory CD8+ T cell responses against identified ligands in ORFV-immunized mice including virus dose and route of administration [59,60], antigen presentation pattern in infected HeLa-K^b^ cells and possible requirements for posttranslational modifications of ORFV epitopes [43,61]. Despite the nonimmunogenic H-2K^b^ ligands identified, all these peptides are naturally presented on MHC molecules, providing evidence for epitope screening. Further infection of other cell lines expressing distinct MHC haplotypes will enable broader insight into the repertoire CD8+ T cell epitopes of ORFV. Considering the importance of CD4+ T cells in the control of viral replication, characterization of MHC class II ORFV ligands would be of interest.

In contrast to the lack of CTL immunity against ORFV-derived peptides, strong multifunctional transgene-specific CD8+ T cell response was elicited by the ORFV after two intramuscular immunizations. Anti-ORFV antibodies are known to play no or very little role in protection against infection as the passive antibody transfer does not confer protection against a virus challenge [4,21]. Moreover, no correlation between serum antibody titers and severity of viral lesions has been described for ORFV [4,21]. Lack of neutralizing antibodies could be linked to the action of the virus immunomodulatory proteins interfering with the host immune response to avoid virus elimination [21,62,63]. Another reason could be the fact that neutralizing targets are complex and comprised of multisubunit structures, as it has been described for VACV [64]. However, the absence of neutralizing antibodies and negligible ORFV-specific cellular immunity enable multiple vaccinations and make ORFV an attractive vaccine platform for infectious diseases and cancer. In addition, the lack of the memory CD8+ T lymphocytes against ORFV epitopes facilitates efficient priming and further enhancement of the antigen-specific responses after repeated immunizations or heterologous prime-boost regimens. Consequently, the ORFV vector strain D1701-V represents a potent carrier for the development of efficient vaccines.

## 5. Conclusions

In this study we investigated the induction of transgene- and vector-specific CD8+ T lymphocyte responses by the ORFV strain D1701-V in mice. We applied a MHC ligandome approach to generate a comprehensive list of ORFV-specific epitopes that might be presented to CD8+ T cells during immunization. For the first time we report the identification of 36 unique D1701-V ORFV-derived H-2K^b^ peptides. Detailed immunogenicity analysis proved that none of the evaluated virus epitopes is immunogenic in mice, but a robust CD8+ T cell immunity is induced against the inserted foreign antigen. These findings reinforce that ORFV strain D1701-V represents an effective and safe recombinant vector for therapeutic and prophylactic vaccines. Additionally, the experiments describe prerequisites for the selection of T cell epitopes exploitable for the generation of ORFV-based vaccines by reverse genetics.

## Figures and Tables

**Figure 1 vaccines-08-00295-f001:**
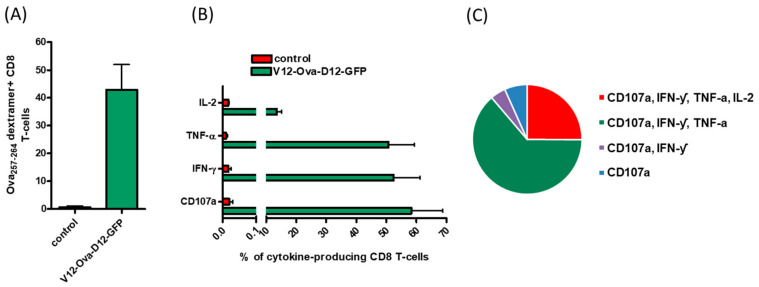
Transgene-specific CD8+ T cell response induced by the ORFV vector. H-2K^b^ C57BL/6 mice (*n* = 6) were immunized i.m. two times with V12-Ova-D12-GFP or negative control V-D12-mCherry. Ova-specific CD8+ T cell response in individual mice was determined one week after the second administration. (**A**) Frequency of specific cytotoxic T lymphocytes of the total CD8+ T cells in the spleen was assessed by Ova_257-264_ dextramer staining. (**B**) Percentage of Ova_257-264_ SIINFEKL peptide-specific CD8+ T cells producing the indicated cytokines was determined by intracellular cytokine staining. (**C**) Pie chart shows the extent of simultaneous CD107a, TNF-α, IFN-γ and IL-2 production by the Ova-specific CD8+ T cells. Frequencies are shown as means ± SEM. Ova, ovalbumin; TNF-α, tumor necrosis factor alpha; IFN-γ, interferon-gamma; IL-2, interleukin-2.

**Figure 2 vaccines-08-00295-f002:**
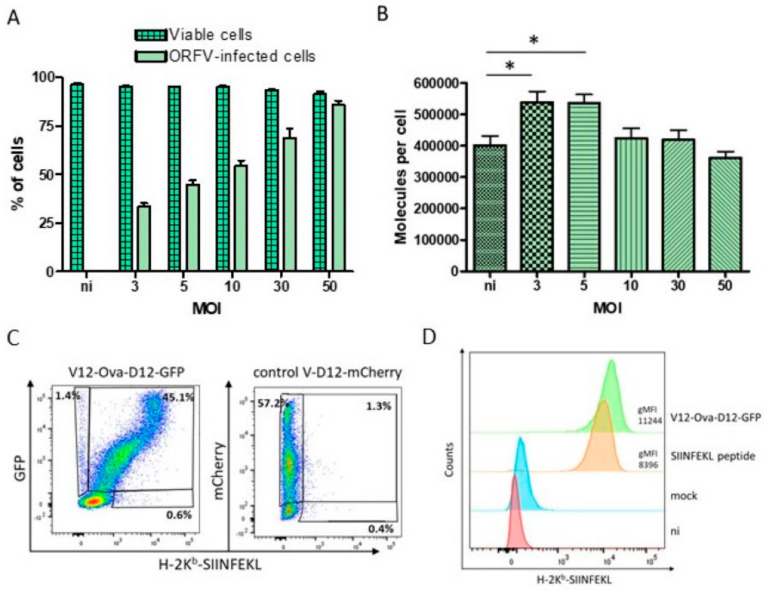
HeLa-K^b^ represent target cells for the identification of ORFV-derived major histocompatibility complex (MHC) class I peptides. (**A**) Cell viability and infection rate following 20 h exposure to ORFV. HeLa-K^b^ cells were infected with indicated multiplicity of infection (MOI). Percentages of viable cells after staining with Zombie Aqua dye and infected cells by mCherry expression were determined by flow cytometry. The data represent the means ± SEM for three independent experiments. (**B**) H-2K^b^ surface expression after ORFV infection with indicated MOI. Twenty hours post infection absolute numbers of H-2K^b^ molecules on the cell surface were determined by flow cytometry using the quantitative indirect immunofluorescence kit. Data shown are means ± SEM from three replicates. (**C**,**D**) Epitope presentation by HeLa-K^b^ cells. Cells were infected with V12-Ova-D12-GFP or negative control V-D12-mCherry with MOI 5 or loaded with synthetic Ova_257-264_ SIINFEKL peptide. Twenty hours after infection cell surface staining was performed using anti-mouse SIINFEKL bound H-2K^b^ antibody followed by flow cytometry analysis. ORFV infection was determined by GFP or mCherry expression. Data are representative of two independent experiments. * *p* < 0.05; ni, non-infected cells; gMFI, geometric mean fluorescence intensity; Ova, ovalbumin.

**Figure 3 vaccines-08-00295-f003:**
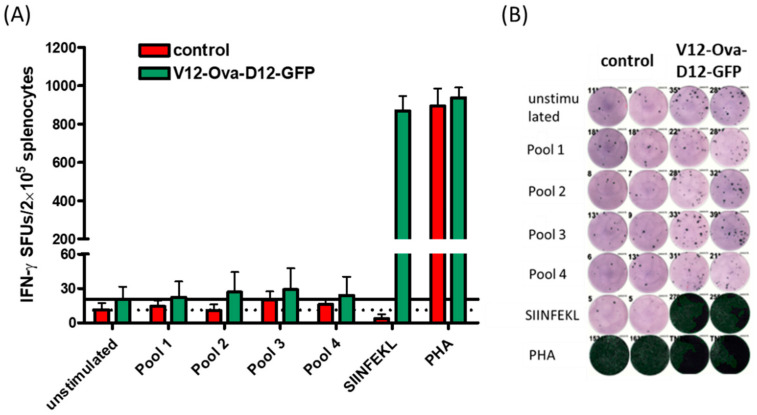
Peptide-specific T cell responses by IFN-ƴ ELISPOT in ORFV immunized mice. H-2K^b^ C57BL/6 mice (*n* = 6) were immunized twice with V12-Ova-D12-GFP or control ORFV by i.m. route. Ova_257-264_ SIINFEKL- and ORFV-derived H-2K^b^ peptide-specific responses were determined seven days after the second administration. (**A**) Numbers of IFN-ƴ spot forming units (SFUs) after stimulation of splenocytes from individual mice with synthetic peptide pools (Pools 1–4) or Ova_257-264_ SIINFEKL-peptide. Spot counts of >1000 were set to 1000 because of inaccurate spot count due to technical limitations. Dotted and solid lines indicate the mean SFUs in negative control wells within control ORFV or V12-Ova-D12-GFP group respectively. Data are shown as means ± SEM of all analysed samples in two independent technical replicates. (**B**) Representative IFN-ƴ ELISPOT assay showing splenocyte responses of control ORFV or V12-Ova-D12-GFP immunized mice. PHA, phytohemagglutinin; Ova, ovalbumin; IFN-γ, interferon-gamma.

**Figure 4 vaccines-08-00295-f004:**
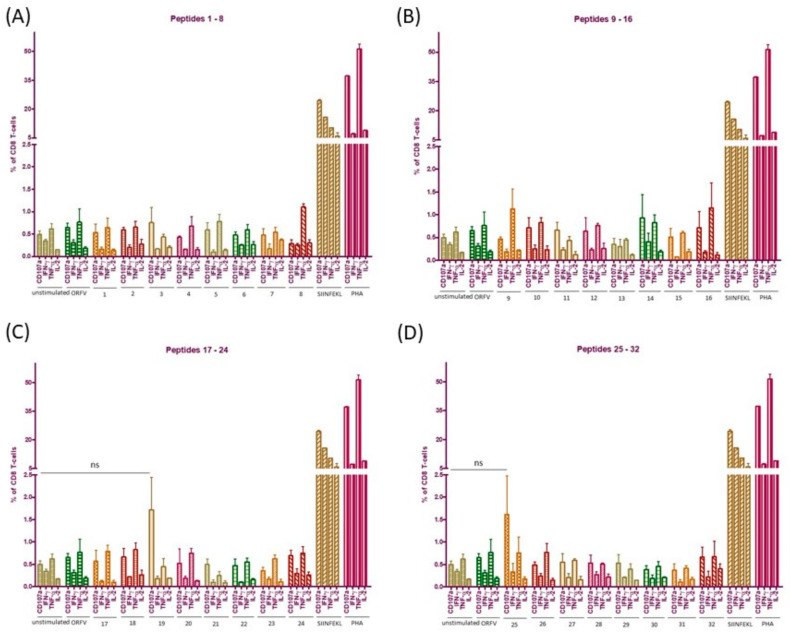
CD8+ T cell responses to the individual ORFV-derived peptides by intracellular cytokine staining (ICS) in ORFV immunized mice. H-2K^b^ C57BL/6 mice (*n* = 6) were immunized twice with V12-Ova-D12-GFP or control ORFV by i.m. route. One week after the second administration specific cytotoxic T lymphocyte responses to the individual ORFV-derived peptides within pooled splenocytes were evaluated using ICS. Percentages of CD107a, TNF-α, IFN-γ and IL-2 expressing CD8+ T cells after stimulation with synthetic peptides (**A**) 1–8, (**B**) 9–16, (**C**) 17–24 and (**D**) 25–32, control ORFV or Ova_257-264_ SIINFEKL peptide. Data are shown as means ± SEM of three independent technical replicates. PHA, Phytohemagglutinin; ns, not significant; Ova, ovalbumin; TNF-α, tumor necrosis factor alpha; IFN-γ, interferon-gamma; IL-2, interleukin-2.

**Table 1 vaccines-08-00295-t001:** H-2K^b^-associated peptides derived from the D1701-V *Orf virus* (ORFV).

No.	Sequence	Length (aa)	UniProt Accession No.	Antigen/Protein *	Position	NetMHC Binding Affinity (nM)	NetMHC % Rank of Predicted Affinity
1	AIYGFGVTF	9	ADY76886.1; ADY76885.1	ORFV056 RNA-polymerase subunit RPO147	569–577	248.6	0.5757
2	VGYPRQNAV	9	ADY76846.1; ADY76844.1	ORFV097 DNA-polymerase processivity factor	364–372	95.7	0.2477
3	IGYMVKNL	8	ADY76734.1	ORFV075 Rifampicin resistance protein	212–219	19.1	0.0441
4	VGFVHPIAM	9	ADY76731.1	ORF079 Virion core protein P4b	258–266	51.9	0.1255
5	SILKFEERL	9	ADY76728.1	ORF083 Early transcription factor VETFL	286–294	93.8	0.2427
6	LNLMYPNI	8	ADY76725.1	ORFV86 Virion core precursor protein P4a	172–179	26.7	0.0674
7	SGSVPYARL	9	ADY76722.1	ORFV090 IMV membrane protein	73–81	140.6	0.3490
8	AAFEFRDL	8	ADY76890.1	ORF052 putative IMV membrane protein	72–79	12.6	0.0253
9	VGFVHPIA	8	ADY76731.1	ORF079 Virion core protein P4b	258–265	485.9	0.9853
10	KILAPFNFL	9	ADY76886.1; ADY76885.1	ORFV056 RNA-polymerase subunit RPO147	834–842	385.2	0.8315
11	ISALFKQL	8	ADY76725.1; ADY76726.1	ORFV086 Virion core protein P4a	781–788	14.2	0.0296
12	TEFPVFEEL	9	ADY76883.1	ORFV058 (IMV, viral entry)	182–190	3095.8	4.6680
13	VNIVRQEEL	9	ADY76792.1	ORFV016 Unknown	35–43	270.1	0.6179
14	AIIKYTDL	8	ADY76837.1	ORFV110 EEV glycoprotein	37–44	89.3	0.2306
15	SILERYNLF	9	ADY76725.1; ADY76726.1	ORFV086 Virion core protein P4a	810–818	130.5	0.3285
16	VFFRVTVL	8	ADY76779.1	ORFV028 DNA-binding protein	644–651	73.4	0.1860
17	GSVPYARL	8	ADY76722.1	ORFV090 IMV membrane protein	74–81	55.2	0.1387
18	QNYSYSERLL	10	ADY76809.1	ORFV129 Ankyrin/F-box protein	115–124	180.9	0.4297
19	RVNTFTAV	8	ADY76793.1	ORFV015 Unknown	33–40	738.0	1.4130
20	TAVDFTQFL	9	ADY76778.1	ORFV029 Unknown	244–252	172.0	0.4073
21	QNYSYSERL	9	ADY76809.1	ORFV129 Ankyrin/F-box protein	115–123	24.3	0.0592
22	AIYAFRLT	8	ADY76719.1	ORFV094 Phosphorylated IMV membrane protein	160–167	150.9	0.3685
23	ANVDFMEYV	9	ADY76886.1; ADY76885.1	ORFV056 RNA-polymerase subunit RPO147	880–888	397.0	0.8488
24	ISVMFNNV	8	ADY76728.1	ORF083 Early transcription factor VETFL	484–491	10.5	0.0198
25	VIFGRQPSL	9	ADY76886.1; ADY76885.1	ORFV056 RNA-polymerase subunit RPO147	374–382	59.9	0.1538
26	EQFSFSNV	8	ADY76841.1	ORFV101 RNA-polymerase subunit RPO132	623–630	259.0	0.5952
27	LIREFANL	8	ADY76728.1	ORF083 Early transcription factor VETFL	294–301	19.5	0.0456
28	IAPQLRSL	8	ADY76724.1	ORF088 Virion core protein	24–31	418.0	0.8785
29	ISIPRSVGF	9	ADY76841.1	ORFV101 RNA-polymerase subunit RPO132	427–435	195.8	0.4646
30	SIAPMNTGF	9	ADY76778.1	ORFV029 Unknown	216–224	3273.7	4.8979
31	IAPQLRSLL	9	ADY76724.1	ORF088 Virion core protein	24–32	331.2	0.7371
32	SATQFQSV	8	ADY76731.1	ORF079 Virion core protein P4b	230–237	482.6	0.9800
33	SFVVVAQI	8	ADY76769.1	ORFV040 Glutaredoxin-like protein	98–105	2254.5	3.5474
34	SIVSFKPTL	9	ADY76731.1	ORF079 Virion core protein P4b	386–394	75.0	0.1902
35	TNVEIGKL	8	ADY76729.1	ORFV082 Unknown	277–284	1825.4	2.9640
36	VIEIFKQL	8	ADY76732.1	ORFV077 Late transcription factor VLTF-3	130–137	216.0	0.5087

Note: * ORFV open reading frames with the functions inferred from the sequence homologies to VACV.

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
