# Peer review of "Orf Virus-Based Vaccine Vector D1701-V Induces Strong CD8+ T Cell Response against the Transgene but Not against ORFV-Derived Epitopes"

_vaccines, 2020, doi:10.3390/vaccines8020295_

Round 1
Reviewer 1 Report
The manuscript is interesting although there are several issues that must be improved. Even though the system is simple and the methods are adequate, I do not think that choosing the peptides based on H-2Kb was the best option. In fact figure 2 C there is a heterogeneous amount of cells shown in the histogram. A reasonable approach would have been to assess the response to the peptides with the highest affinity. The analysis of figure 4 and S1 should have been easier. The values of IFN gamma in most cases is lower than the negative control, how do the authors explain this event? In addition, in the positive controls, the values are lower than expected. The authors claim no significant increase in CD107a with two peptides 19 and 25, why this was not further analyzed? In fact, it is remarkable that the levels of CD107a were lower than expected in figure S1 based on figure 4. I would suggest the authors to do sorting on the two clear subpopulations of cells observed on figure 2C and then redo the assays for both. Moreover, the discussion should be enhanced to discuss the importance of the finding.
Author Response
Comment “I do not think that choosing the peptides based on H-2Kb was the best option”:
We agree that the used model has some limitations as not reflecting the natural situation completely. Nevertheless, it is well-suitable and relevant to test our hypothesis. The presentation of epitopes on MHC molecules denotes the prerequisite for a cellular immune response. The used inbred mouse C57BL/6 expresses the MHC class I haplotype H-2Kb. Consequently, peptides bound to H-2Kb after infecting the H-2Kb-positive cell line (HeLa-Kb) enabled us to elute and identify the most abundant ORFV-derived epitopes. To make that point clearer and for better explanation of the chosen experimental setup, we now inserted in “Results” lines 233 and 263 this information.
Comment “In figure 2C there is a heterogeneous amount of cells shown in the histogram”:
Before we performed identification of ORFV-derived H-2Kb-presented peptides by LC-MS/MS we aimed to validate the ability of HeLa-Kb cell line to present ORFV-derived epitopes. Therefore, we infected HeLa-Kb cells with V12-Ova-D12-GFP or negative control V-D12-Cherry with MOI 5 resulting in approx. 50 % of infected cells which is in line with Figure 2A. Owing to a commercially available antibody, we were able to specifically detect Ovalbumin-derived epitope SIINFEKL bound to H-2Kb. We stained V12-Ova-D12-GFP and V-D12-Cherry infected HeLa-Kb cells and showed that nearly all V12-Ova-D12-GFP infected cells enable Ova257-264 SIINFEKL peptide processing and presentation (GFP+ SIINFEKL+) in comparison to non-infected cells (GFP- SIINFEKL-) or V-D12-Cherry infected cells. Therefore, we believe that nearly all ORFV-infected HeLa-Kb cells present H-2Kb bound ORFV-derived peptides and heterogeneity is minimal. Those results are described in the manuscript lines 297 – 302. However, we prepared a new Figure 2C with minor changes that includes some more detailed information about the detected populations. We would leave the decision to the editor if he believes replacing the Figure 2C is necessary.
Comment “A reasonable approach would have been to assess the response to the peptides with the highest affinity”:
We assume that the reviewer's comment refers to Figures 2C and 2D. We agree that it would be of course most desirable to assess the response also to other peptides than SIINFEKL. However, no antibodies are available to detect ORFV-derived epitopes in the context of H-2Kb. To make that point clearer, we now inserted in “Discussion” lines 446 - 448 this information.
Regarding the ligandome analysis, we can be confident having identified most, if not all, of the H-2Kb peptides with the highest affinity for MHC class I complexes, sufficient to be presented. Thus, we did not go into more detail to that comment.
Comment “The values of IFN gamma in most cases is lower than the negative control”:
We have previously observed that background IFN-γ responses in individual mice after immunization with ORFV range between 0.1% and 0.4% of all CD8+ T cells. As all CD8+ T cells originate from the identical animals, we assume that the somewhat higher IFN-γ value in the negative control happened by chance.
Comment “In the positive controls, the values are lower than expected”:
We think that relatively lower values of IFN-γ and IL-2 in ex vivo ICS might have been influenced by high values of TNF-α and CD107a after PHA stimulation.
Comment “No significant increase in CD107a with two peptides 19 and 25, why this was not further analyzed”:
Due to the higher numbers of CD107a+ CD8+ T cells in response to peptides 19 and 25 detected in ex vivo ICS we decided to further investigate peptide 19 and 25. Thus, we performed a 14 days splenocyte re-stimulation with the respective ORFV peptides (shown in Fig S1). Whenever specific cells are present among the splenocytes, peptide re-stimulation would lead to their expansion excluding priming of naïve T cells. However, we didn’t observe any positive responses against these peptides after 14 days. Moreover, we used this assay for all other identified ORFV peptides to exclude the possibility that low-frequency memory CD8+ T cells might be below detection limits of ex vivo ICS.
Nevertheless, we realize that it is obviously difficult to follow our approach. Therefore, we rephrased lines 413-414 to emphasize the experimental proof and provided in the “Results” lines 416 - 420 this information.
Comment “it is remarkable that the levels of CD107a were lower than expected in figure S1 based on figure 4”:
After 14 days re-stimulation of pooled splenocytes with the ORFV peptides, if no memory CD8+ T-cells have been re-activated and expanded, CD107a expression is comparable to the unstimulated control. Percentage of CD107a+ cells in ex vivo ICS and after 14 days re-stimulation is different due to different cell culture conditions as well as protocol applied. To clarify this point, we inserted in “Results” lines 417 - 418 and 423 – 425 this information.
Comment “I would suggest the authors to do sorting on the two clear subpopulations of cells observed on figure 2C and then redo the assays for both”:
We were not completely sure if we understand the reviewer’s suggestion properly or if this comment became superfluous after the changes in Figure 2C. However, we would not like to perform such an assay by mixing a human cell line with primary murine cells for investigation of the T-cell response. Thus, we have not taken up that comment further in the text.
Comment “The discussion should be enhanced to discuss the importance of the finding”:
We agree with the Reviewer that a discussion of the importance of the finding could be enlarged. Therefore, we now revised "Discussion" lines 481 – 488 by including the following:
Anti-ORFV antibodies are known to play no or very little role in protection against infection as the passive antibody transfer does not confer protection against a virus challenge (Haig and Mercer 1998; Lloyd et al, 2000). Moreover, no correlation between serum antibody titers and severity of viral lesions has been described for the ORFV (Haig and Mercer 1998; Lloyd et al, 2000). Lack of neutralizing antibodies could be linked to the action of the virus immunomodulatory proteins interfering with the host immune response to avoid virus elimination (Lloyd et al, 2000; Brun, 2003; Haig 2001). Another reason could be the fact that neutralizing targets are complex and comprised of multisubunit structures, as it has been described for VACV (Maruri-Avidal et al, 2011).
Reviewer 2 Report
The short-lived ORFV-specific immune response and the absence of virus neutralizing antibodies (a T-dependent immune response) is an interesting immune issue. Is it true in humans? Here authors have demonstrated that unlike 36 ORFV derived MHC-I peptides, immunizaion with ovalbumin-expressing ORFV elicited a strong SIINFEKL-specific CD8+T response. Based on some interesting experiments, authors have concluded ORFV strain D1701 is an attractive vector for vaccine development. I have following issues with this manuscript that must be answered in the revised manuscript.
- Describe the genotype of D1701-V-D12-mCherry and D1701-V12-Ova-D12-GFP viruses, including genes deleted/inserted, in materials and methods. It is important to see the validity of controls in different experiments. It is important for readers not familiar with the ORFV immunology.
- Similarly, strong transgene responses and negligible anti-ORFV immunity, and absense of systemic virus spread in immuno-deficient animals appear very attractive, but most readers would like to see this issue addressed in the Discussion. Possible mechanisms? Is it immune-tolernace or peudo-tolerance?
- Describing Penicillin-Stretomycin as 1% (lines 87 and 89) does not make sense. It should be in micrograms or units, as necessary.
- Media (lines 97 and 99 etc.) should be "medium".
- Sentences should not start with numbers (lines 165 and 202 etc.).
Author Response
Comment “Is it true in humans?”:
Zoonotic transmission of wild-type ORFV from animals to humans has been reported, however ORFV-specific immune response in humans has not been investigated in detail. The ORFV lesions generally resolve in short times suggesting that a human immune response is very similar to that of sheep.
We agree that a description of ORFV-specific immune responses in humans is missing. Therefore, we now inserted in the “Introduction” lines 53 – 54 the following: Analyses of human patients accidentally infected with wild-type ORFV, whose lesions generally resolve in short times, suspected a human immune response very similar to that of sheep (Yirrell et al., 1994).
Comment 1 “Describe the genotype of D1701-V-D12-mCherry and D1701-V12-Ova-D12-GFP viruses, including genes deleted/inserted, in materials and methods.”:
We agree that a description of the Ova-expressing ORFV recombinant is missing. Therefore, we now revised Chapter 2.1 "Cells and viruses" lines 99 – 104 by including the following:
The mouse Ova gene was synthesized (Gene Art, Thermo Fisher Scientific) and cloned as a HindIII-EcoRI fragment into plasmid pV12-mCherry (Rziha et al., 2019). The resulting transfer plasmid pV12-mOva was used for transfection of Vero cells infected with D1701-V12-mCherry-D12-GFP (Rziha et al., 2019) to replace the mCherry gene by the Ova gene. Ova-specific PCR allowed selection and purification of the Ova-expressing recombinant D1701-V12-Ova-D12-GFP essentially as described earlier (Rziha et al., 2019).
Moreover, we revised "Introduction" lines 38 – 41 by including the following:
Importantly, different derivatives of the Vero cell culture adapted vector virus strain D1701-V were apathogenic even after high dose infection of immunosuppressed sheep (Rziha et al., 2000). Whether and which of the ORFV genes deleted in D1701-V remains to be clarified (Rziha et al., 2019).
Description of the construction including genomic map of D1701-V-D12-mCherry has been published recently in Rziha et al., 2019, and thus we propose the citation of this publication should be sufficient.
Comment 2 "Strong transgene responses and negligible anti-ORFV immunity, and absence of systemic virus spread in immuno-deficient animals appear very attractive, but most readers would like to see this issue addressed in the Discussion”:
We agree with the Reviewer that a description of possible mechanisms is important and missing in the manuscript. Therefore, we now revised "Discussion" lines 481 – 488 by including the following:
Anti-ORFV antibodies are known to play no or very little role in protection against infection as the passive antibody transfer does not confer protection against a virus challenge (Haig and Mercer 1998; Lloyd et al, 2000). Moreover, no correlation between serum antibody titers and severity of viral lesions has been described for the ORFV (Haig and Mercer 1998; Lloyd et al, 2000). Lack of neutralizing antibodies could be linked to the action of the virus immunomodulatory proteins interfering with the host immune response to avoid virus elimination (Lloyd et al, 2000; Brun, 2003; Haig 2001). Another reason could be the fact that neutralizing targets are complex and comprised of multisubunit structures, as it has been described for VACV (Maruri-Avidal et al, 2011).
Comment 3 “Describing Penicillin-Stretomycin as 1% (lines 87 and 89) does not make sense”:
We agree with the Reviewer. Therefore, we now revised Chapter 2.1 "Cells and viruses" lines 94 and 97 by including the following: 50 U/ml Penicillin and 50 μg/ml -Streptomycin.
Comment 4 “Media (lines 97 and 99 etc.) should be "medium":
We agree with the Reviewer and now revised Chapter 2.1 "Cells and viruses" lines 109 and 112 by changing “media” to “medium”.
Comment 5: “ Sentences should not start with numbers (lines 165 and 202 etc.)
We agree with the Reviewer. Therefore, we now revised Chapter 2.8 "IFN-γ ELISPOT assay" line 178 and Chapter 2.11 "In vitro re-stimulation of splenocytes" line 215 by adding “In total”.

Round 2
Reviewer 1 Report
The manuscript was improved. Some minor issues remained like the flow cytometry assay figure 2C